# Can Corrole Dimers Be Good Photosensitizers to Kill Bacteria?

**DOI:** 10.3390/microorganisms10061167

**Published:** 2022-06-07

**Authors:** Paula S. S. Lacerda, Maria Bartolomeu, Ana T. P. C. Gomes, Ana S. Duarte, Adelaide Almeida, Maria A. F. Faustino, Maria G. P. M. S. Neves, Joana F. B. Barata

**Affiliations:** 1CESAM, University of Aveiro, 3810-193 Aveiro, Portugal; placerda@ua.pt (P.S.S.L.); maria.bartolomeu@ua.pt (M.B.); aalmeida@ua.pt (A.A.); 2LAQV-REQUIMTE and Department of Chemistry, University of Aveiro, 3810-193 Aveiro, Portugal; faustino@ua.pt (M.A.F.F.); gneves@ua.pt (M.G.P.M.S.N.); 3Department of Biology, University of Aveiro, 3810-193 Aveiro, Portugal; 4Universidade Católica Portuguesa, Faculty of Dental Medicine (FMD), Center for Interdisciplinary Research in Health (CIIS), 3504-505 Viseu, Portugal; apgomes@ucp.pt (A.T.P.C.G.); asduarte@ucp.pt (A.S.D.)

**Keywords:** corroles, corrole dimers, photosensitizers, antimicrobial photodynamic therapy, *Staphylococcus aureus*, uptake, cytotoxicity, Vero cells

## Abstract

Corroles possess key photophysical and photochemical properties to be exploited as therapeutic agents in antimicrobial photodynamic therapy (aPDT). Herein, we present for the first time the antimicrobial efficiency of three corrole dimers and of the corresponding precursor against the Gram(+) bacterium *Staphylococcus aureus*. Additionally, to explore future clinical applications, the cytotoxicity of the most promising derivatives towards Vero cells was evaluated. The aPDT assays performed under white light irradiation (50 mW/cm^2^; light dose 450 J/cm^2^) and at a corrole concentration of 15 µM showed that some dimers were able to reduce 99.9999% of *S. aureus* strain (decrease of 5 log_10_ CFU/mL) and their photodynamic efficiency was dependent on position, type of linkage, and aggregation behavior. Under the same light conditions, the corrole precursor **1** demonstrated notable photodynamic efficiency, achieving total photoinactivation (>8.0 log_10_ CFU/mL reduction) after the same period of irradiation (light dose 450 J/cm^2^). No cytotoxicity was observed when Vero cells were exposed to corrole **1** and dimer **3** for 24 h according to ISO guidelines (ISO 10993-5) for in vitro cytotoxicity of medical devices. The results show that corrole dimers, dependent on their structures, can be considered good photosensitizers to kill *Staphylococcus aureus*.

## 1. Introduction

Corroles are aromatic contracted tetrapyrrolic macrocycles with unique physicochemical features to be used in different fields including oxidative catalysis, sensing, and therapeutic applications [1,2,3,4,5]. Under the context of medicinal applications, their optical properties such as high extinction molar coefficients, fluorescence quantum yield (Φ_Flu_), and ability to generate singlet oxygen (^1^O_2_) make them obvious candidates to be exploited as therapeutic agents, namely for photodynamic therapy (PDT) for cancer [5,6,7,8,9,10,11,12,13,14] and antimicrobial photodynamic therapy (aPDT) approaches [15,16]. The principles of PDT for treating cancer cells and aPDT for eliminating microorganisms are the same; both approaches require the activation of a dye (the photosensitizer, PS) by an adequate visible light source in the presence of dioxygen (^3^O_2_) in order to produce reactive oxygen species (ROS, e.g., ^1^O_2_, among others) responsible for selective killing. Recent developments show that aPDT can be considered a promising alternative to antibiotics in order to deal with the emerging public health problem associated with the growing rate of resistant bacteria strains [17,18,19]. In fact, a high number of in vivo and in vitro studies have confirmed the high efficiency of aPDT in photoinactivating a broad spectrum of pathogens [20]. Most of this success is related to the accessibility of PSs with adequate structural features. Among the different types of PSs (e.g., phenothiazinium dyes, psoralens, perylenequinonoid pigments, fullerenes), the antimicrobial achievements obtained with natural and synthetic porphyrin derivatives and analogues are particularly promising [21,22]. However, when compared with other porphyrinoids the use of corroles in aPDT is much more limited. Cationic corrole derivatives have already proved efficient antimicrobial activity against the Gram(−) bacterium *Aliivibrio fischeri* [15] and fungi species *Aspergillus niger*, *Cladosporium cladosporoides*, and *Penicillium purpurgenum*, even with observed damages on spore forms [16]. Negatively charged corrole derivatives have also demonstrated photoinactivation efficiency against mold fungi species [16]. However, the activity of non-charged (electrically neutral) corrole derivatives against Gram(+) bacteria has not been explored as far as we know. 

Considering our interest in developing new PSs, namely of the corrole type with efficient antimicrobial activity, we hypothesized the possibility of using corrole dimers (Figure 1) as new antimicrobial PS agents [23,24]. In terms of the key photophysical and photochemical characteristics to generate ROS, in particular singlet oxygen, thermodynamic stability, and non-dark toxicity required for a compound to be used as PSs, the electronic absorption of the corrole dimers is particularly promising since they cover almost all the visible range of the electromagnetic spectrum, which allows the use of non-expensive white light sources to excite the PS, namely light-emitting diodes (LED). The previous approaches to prepare free base corrole dimers involved: (a) a multistep process that relies on regiospecific palladium-catalyzed oxidative coupling reactions [25] and (b) heating of corrole monomer under a strict inert atmosphere [26] or under an oxidative atmosphere [27]. Herein, the dimerization of gallium complex of 5,10,15-tris(pentafluorophenyl)corrole was mediated by acids using a mixture of common laboratory acids.

In this paper, we present for the first time the antimicrobial efficiency of corrole dimers against the Gram(+) bacterium *Staphylococcus aureus*. *S. aureus* is an important and well-known opportunistic pathogen [28,29] that produces and excretes a variety of virulence factors whose function is to disrupt, invade, and override the host immune system [29,30]. Although conventional antibiotic therapy may be effective in some cases of *S. aureus* infection, the increasing prevalence of multidrug resistant strains of this bacterium has led to treatment failure and, consequently, an increased mortality ratio [31]. *S. aureus* is capable of high mutation rates and when beneficial to the survival of the bacterial cell under selective pressure, the mutations accumulate in the bacterial genome and the spread of the variant strains occurs [28]. Additionally, to investigate the possibility of future clinical application of these compounds as antibacterial agents, the cytotoxicity of the most promising tested compounds was evaluated towards Vero cells.

## 2. Materials and Methods

### 2.1. Photosensitizer

The corrole precursor 5,10,15-tris(pentafluorophenyl)corrolatogallium(III)(pyridine) (**1**) was synthesized according to literature [32,33]. The general procedure for the synthesis of corrole dimers **2**–**4** (Figure 1), as well as their structural characterization, are available in Appendix A. The spectrophotometric and spectrofluorimetric measurements as well as the determination of singlet oxygen production are also available in Appendix A, respectively.

### 2.2. Biological Assays of Corrole PSs against S. aureus

The photodynamic efficiency of each corrole dimer **2**, **3** and **4** and also of the corrole precursor **1** was evaluated against the Gram(+) bacterium *S. aureus.* All the aPDT assays were performed using the same PS concentration (15 µM) and three independent experiments with two replicates each.

#### 2.2.1. Bacterial Strain and Growth Conditions

Stocks of the bacterium were stored at −80 °C in 10% glycerol. Before each assay, a bacterial glycerol stock was inoculated in 30 mL of fresh Tryptic Soy Broth (TSB; Liofilchem, Roseto degli Abruzzi, Italy) medium, and grown overnight at 37 °C, with orbital shaking (120 rpm). An aliquot of the grown bacterial culture was inoculated on fresh TSB medium and incubated as described above until the stationary phase (the less susceptible growth phase where the bacterial metabolic processes are slower [34,35,36]) of the bacterial culture was reached; at this stage, the bacterial concentration presented an optical density (OD) of ca. 1.6 at 600 nm, corresponding to a bacterial cell concentration of ca. 10^9^ colony-forming units per mL (CFU/mL). This procedure was performed before each aPDT assay.

#### 2.2.2. Stock Solutions of the Photosensitizers

Stock solutions of the corrole dimers **2**, **3**, **4**, and of precursor **1** used in the aPDT studies were prepared in dimethylsulfoxide (DMSO) at a concentration of 500 µM and stored refrigerated (4 °C) protected from light. Before each assay, the stock solutions were kept at room temperature until thawing and then sonicated for 15 min to ensure the homogeneity of the solution. The appropriated volume of each stock solution was transferred to the prepared bacterial suspension to reach the final concentration of 15 µM.

#### 2.2.3. Light Source Conditions

The irradiation system used in the aPDT assays was composed by a low energy consuming white light-emitting diode (LED) (400–700 nm, 1400 lm, 5500 K) (EL^®^MARK Holding, Varna, Bulgaria). Light irradiance was measured with a power meter FieldMaxII-TOP combined with a high-sensitivity thermopile sensor PS19Q (Coherent, Santa Clara, CA, USA) and adjusted to 50 mW/cm^2^ before each assay; the distance between the light source and the sample was around 3.0 cm.

#### 2.2.4. aPDT Experimental Setup

All the aPDT assays were performed at the same PS concentration (15 µM). The concentrations of the PSs were chosen according to the previously performed toxicity tests of the solvent (DMSO) considering the concentration of the PSs stock solutions (500 µM) (Appendix A).

Before each aPDT assay, a portion of the grown bacterial culture was ten-fold diluted in phosphate-buffered solution (PBS) (reaching a bacterial concentration of ca. 10^8^ CFU/mL, corresponding to an OD of ca. 0.7 at 600 nm), and then an adequate volume of each PS stock solution was added. In order to promote the homogeneity of the suspension, and the interaction of the PS molecules with the bacterial cells, each sample was maintained for 15 min in the dark under magnetic stirring. After this incubation period, the samples were irradiated under white light for a total of 180 min at an irradiance of 50 mW/cm^2^. Along with the samples, light and dark controls were also performed. Light controls containing only bacterial suspension of *S. aureus* in PBS were irradiated in order to evaluate if the irradiation conditions alone would have any antimicrobial effect; dark controls, also containing the bacterial suspension in PBS and the PS at the same concentration as in the samples (15 µM) were maintained in the dark to evaluate if the PS in the dark had some toxic effect against the bacterial cells.

After defined treatment periods (0, 30, 60, 90, 120, 150, and 180 min), aliquots of the samples and the controls were collected and serially diluted in PBS. Then, aliquots from the diluted suspensions were drop-plated in Petri dishes previously prepared with Tryptic Soy Agar (TSA; Liofilchem, Roseto degli Abruzzi, Italy), and incubated at 37 °C, for 24 h. The bacterial colony forming units (CFU) were counted and the results of the bacterial concentrations were expressed as log_10_ (CFU/mL). The detection limit of the used method is 2.0 log CFU/mL.

#### 2.2.5. Photosensitizer Binding

A bacterial suspension (~10^8^ cells/mL) was incubated for 15 min in the dark at room temperature in the presence of each PS at the same concentration used in the bacterial inactivation studies (15 µM). The unbound PS was removed from the suspension by centrifugation at 13,000× *g* for 5 min (1730R ©, Gyrozen Co., Ltd., Gimpo, Korea). The obtained pellets were washed three times with PBS and then centrifuged at the same conditions. The pellets were then re-suspended in 100 µL of DMSO for digestion and vigorously shaken with a vortex. The concentration of each PS in the digested extracts was determined by fluorescence for corrole **1** and by UV-Vis for dimers **2**–**4**. The samples of corrole **1** analyzed by fluorescence were excited at 422 nm. A calibration plot was built using known concentrations of each PS and the fluorescence intensity of the samples was interpolated to determine the corresponding concentration of the samples. The absorbance spectra of the corrole dimers were monitored in the 350–800 nm range. The measured absorbance intensity of known concentrations of corrole dimers in the same digestion solution allowed the building of a calibration plot and then the determination of the corresponding concentration of the samples.

Parallel aliquots of the cell suspension incubated in the presence of the PS were serially diluted and drop plated for the determination of the concentration of viable *S. aureus* cells (CFU/mL). The number of PS molecules/cell was calculated. Three independent assays with two replicates each were performed.

### 2.3. Cytotoxicity Evaluation of Corrole Precursor ***1*** and Dimer ***3***

The study of the cytotoxicity of corrole precursor **1** and dimer **3** efficiency was performed using the Vero cell line (ECACC 88020401, African Green Monkey Kidney cells, GMK clone) according to the International Organization for Standardization (ISO) guidelines (ISO 10993-5) (International Organization for Standardization 2009) [37]. This cell line was cultured in Dulbecco’s Modified Eagle Medium (DMEM, Gibco BRL, Invitrogen) supplemented with 10% (*v*/*v*) of fetal bovine serum (FBS, Gibco BRL, Invitrogen), 100 U/mL penicillin, 100 mg/mL streptomycin, and 0.25 mg/mL amphotericin B (Gibco BRL, Invitrogen).

Vero cells were seeded (9.4 × 10^4^ cells/cm^2^) in 96-well cell culture plates and maintained in culture medium under an air atmosphere containing 5% CO_2_ overnight. The cells were washed twice with PBS and incubated with DMEM solutions of corrole precursor **1** and dimer **3** at 15 μM for 24 h in a humidified incubator with 5% of CO_2_ atmosphere and 95% of air. The viability of Vero cells after incubation with both PSs was determined by measuring the ability of the cells to reduce 7-hydroxy-3H-phenoxazin-3-one-10-oxide sodium salt (Resazurin, Sigma) to resorufin using a microplate reader (SynergyTM HT Multi-Detection Microplate Reader-Biotek^®^) [38]. The data were expressed in percentage of control (i.e., optical density of resorufin (OD 570 nm) from cells not exposed to PSs). Three independent assays with three replicates each were performed.

### 2.4. Statistical Analysis

The significance of the difference in bacterial inactivation among the different compounds was assessed by ANOVA and Tukey’s multiple comparisons test using GraphPad Prism software (GraphPad Software, San Diego, CA, USA). A value of *p* < 0.05 was considered significant. Three independent experiments with two replicates each were performed by each tested condition. For the cytotoxicity studies, statistical significance among the conditions was assessed using the nonparametric Mann–Whitney test. The results are presented as the mean of at least three independent assays with three replicates per assay.

## 3. Results and Discussion

### 3.1. Synthesis and Characterization of Corrole Oligomers

The promising spectroscopic features of the Ga(III) dimers to be used as PSs prompted us to explore a one-step strategy as a synthetic alternative. The possibility that acidic conditions were prone to conducting corrole dimers was found during our studies concerning the hydrolysis of the methyl ester group in gallium(III) complex of 10-(4-methoxycarbonylphenyl)-5,15-bis(pentafluorophenyl)corrole **S1** (see Appendix A) [39].

Following our research on the synthesis of β,β-linked dimers [26,40], we decided to evaluate the potential of the acidic conditions to obtain gallium(III) corrole dimers in just one step, using the more accessible gallium(III) corrole complex **1** as the starting monomeric unit (Figure 1).

The first assay (Table 1, entry 1) was conducted using the previously reported conditions [39]. The solution of corrole **1** in a mixture of acetic acid, trifluoracetic acid and a 5% aqueous solution of sulfuric acid (AcOH/TFA/5% aq H_2_SO_4_) in a 4:2:1 volume ratio was heated at 100 °C. After 2 h, the total consumption of corrole **1**, and the presence of several green compounds were observed. The mass spectrum analysis of this mixture revealed the presence of dimers with different oxidation degrees. After work-up and chromatographic purification, it was possible to isolate three main fractions, which were identified as the doubly linked corrole dimer **2** at 15% yield and the singly linked corrole dimers **3** and **4** at 5% and 14% yields, respectively. Other fractions were also detected, but not in amounts to allow their characterization.

In order to minimize the complexity of the reaction, a second reaction under the same acidic conditions was performed but the reaction time was reduced to 30 min (Table 1, entry 2). Under these conditions, the starting corrole **1** was recovered with 19% yield, and only dimers **4** (8%) and **2** (10%) were isolated in adequate amounts to allow their identification. We also found that under the same acidic conditions, the reduction of the reaction temperature to 40 °C did not bring any positive impact to the reaction performance; most of the starting corrole **1** was recovered after 2 h of reaction and only vestigial amounts of the desired dimers were detected (Table 1, entry 3). These results confirmed that temperature is an important factor for the dimerization type reaction to occur.

In order to evaluate the role of the acids on the reaction profile, some additional experiments were performed in acetic acid (the major acid component) at 100 °C in the absence or in the presence of TFA or H_2_SO_4_ solution (Table 1, entries 4–6).

In 100% of AcOH, corrole **1** was recovered at 61% after 2 h of reaction and the main derivatives were the singly linked corrole dimers **3** and **4**, which were isolated at 10 and 3% yields, respectively; dimer **2** was only detected in a vestigial amount (Table 1, entry 4). The same type of profile was observed in the reactions performed in AcOH and TFA (Table 1, entry 5) although under these conditions dimer **3** was isolated at 5% and dimer **4** at 11%; the starting corrole **1** was recovered at 37% yield. This reaction was stopped after 1 h due to the presence of several decomposition products observed by TLC analysis.

The best conditions to obtain all dimers were found when AcOH and 5% aq. H_2_SO_4_ (6:1) were used (Table 1, entry 6). Under these conditions, corrole **1** was recovered at 56% yield, and dimers **2**, **3**, and **4** were isolated at 12%, 10%, and 15% yields, respectively.

Dimers **2**, **3**, and **4** were characterized by NMR, mass spectrometry, UV-Vis, and fluorescence spectroscopy. The structural characterization obtained for dimers **2** and **3** were in accordance with the literature [27,40].

For dimer **4**, the ^1^H NMR spectrum shows a pattern that corroborates its greater asymmetry when compared to dimer **3** or the highly symmetric dimer **2**. The singlet at 9.7 ppm and a doublet at 9.34 ppm, accompanied by the several duplets due to the resonance of the remaining twelve H-β protons centered at ca. δ 8.9 and δ 8.5 ppm, are compatible with the proposed structure of dimer **4**. The singlet at 9.72 ppm corresponds to the resonance of H-2 (consistent with a substitution at H-3 in a corrole unit). The other expected singlet due to the resonance of H-3′ is attributed as part of the multiplet at 8.72–8.69 ppm (resonance due to a total of four H-β protons), as in the COSY spectrum, only three interactions with three duplets (each due to one proton) are observed for the referred multiplet.

Additionally, the ^19^F NMR spectrum of the dimer **4** also shows an asymmetric profile, with the presence of several multiplets corresponding to the *ortho-*, *para-*, and *meta*-fluorine atoms’ resonances. The mass spectrum shows a peak at *m*/*z* 1724.9 [M + H–2py]^+^ corresponding to the expected protonated molecular ion without the pyridine ligand.

The mechanisms for the synthesis of corrole dimer **2** reported in the literature refer either to a multistep process that relies on regiospecific palladium-catalyzed oxidative coupling reactions or to a thermal oxidative cyclisation. In our case, the mechanism of formation of these corrole dimers is still to be confirmed, but probably involves an oxidative C-C coupling mechanism mediated by the presence of the acids.

### 3.2. Photophysical Properties

The main photophysical features of corrole dimers **2**–**4** such as UV-vis absorption and fluorescence emission properties, Stokes shifts, and Φ_Flu_ are detailed in Appendix A.

The electronic absorption spectra of the synthesized gallium(III) corrole dimers were recorded in DMF (Figure 1). The UV–vis spectra of corrole dimers **3** and **4** revealed the characteristic profile of corrole derivatives, dominated by a Soret-type band at 426 and 425 nm, respectively, and weak and broader Q bands ranging from 550–670 nm [41]. Notice that the Q-band for the corrole dimers **3** and **4** exhibited a red-shift when compared to the corrole precursor **1**. These features are in line with other types of conjugated corrole dimers and followed the same trend as the free base dimeric compounds [26]. The UV-Vis spectrum of corrole dimer **2** was significantly different. In fact, this compound presents the Soret-like band blue-shift at 403 nm when compared to the corrole precursor **1** and a very intense band in the near-infrared region at 723 nm (Figure 1), the result of a π-extended chromophore [27].

The emission behaviors of the obtained corrole dimers **2**, **3**, and **4** were investigated in DMF (Figure 1). The emission of these dimers occurs at longer wavelengths than corrole precursor **1** (λ_em_ 600 nm). The singly linked dimers **4** and **3** showed emission maxima at 656 and 661 nm, respectively. Concerning the doubly linked dimer **2**, a maximum emission at 730 nm was observed, which is significantly shifted towards the red region compared to both precursor and the other dimers due to improved conjugation of this compound, which is in line with literature [27].

Corrole dimers **2**, **3**, and **4** are less emissive molecules (Φ_Flu_ 0.09–0.11) than corrole **1 (**Φ_Flu_ 0.21) [42]. Corrole dimer **2** and precursor **1** showed similar Stokes shift (7 nm vs. 6 nm), corroborating the symmetry and planarity of the dimer [27]. A different outcome was observed for the singly linked dimers **3** and **4**, which showed higher Stokes shift values (40 and 45 nm, respectively), suggesting less symmetrical molecules.

### 3.3. Singlet Oxygen Generation

Considering the possibility of using corrole dimers **2**–**4** as PSs, their ability to generate ^1^O_2_ was evaluated through the decay of 9,10-dimethylanthracene (DMA) under irradiation [43] in the presence of corrole dimers with 5,10,15,20-tetraphenylporphyrin (TPP), a well-known PS, as reference (see Appendix A). The photooxidation of DMA by ^1^O_2_ follows a first order kinetic, as can be confirmed by the data depicted in Figure 2. The results show that corrole dimers **2**, **3**, and **4** are able to generate ^1^O_2_ (Φ_Δ_ 0.21–0.29), although to a lesser extent than the corrole precursor **1** (Φ_Δ_ = 0.69). In the absence of PS, DMA photooxidation was not observed.

Singlet oxygen might not be the only ROS produced during the photodynamic treatment, as it is well documented for tetrapyrrolic derivatives and analogues [44,45]. However, it is recognized that the photodynamic efficiency of this type of compound towards bacterial cells is, in general, strongly dependent on singlet oxygen production and, in its absence, the aPDT treatment would be highly affected. Although the possible intervention of other ROSs cannot be eliminated, considering the similarity of the structures of the dimers, we believe that their contribution would be also similar, as occurs with their production of singlet oxygen.

### 3.4. Biological Assays against Staphylococcus aureus

Although the information in the literature points out the potential use of corroles and their derivatives in aPDT, to date, few studies have been performed in order to assess the efficiency of these compounds against microorganisms, especially dimeric derivatives. The efficiency of corrole **1** and corrole dimers **2**, **3**, and **4** to photoinactivate bacterial strains was evaluated against the Gram(+) *S. aureus* bacterium. In addition to the aPDT assays, the quantification of PS molecules’ adhesion to bacterial cells was also performed in order to understand the differences found in the photoinactivation efficiency between the tested corrole derivatives.

All the aPDT assays were performed under white light irradiation (50 mW/cm^2^) and at the same PS concentration (15 µM). This concentration was chosen after performing, under dark conditions, toxicity tests of the solvent (DMSO) considering different dilutions of the PS stock solutions at 500 µM; the maximum concentration for which the solvent did not present toxicity to *S. aureus* cells corresponded to 15.6 µM (Appendix A). The results summarized in Figure 3 show that all the neutral derivatives exhibit the ability to photoinactivate the Gram(+) bacterium *S. aureus* without demonstrating any dark toxicity.

Under light conditions, the corrole precursor **1** demonstrated notable photoinactivation efficiency when compared with the non-treated bacterial suspension (Anova, *p* < 0.05), achieving total photoinactivation (>8.0 log_10_ CFU/mL reduction) after 150 min of white light irradiation (light dose 450 J/cm^2^) (Figure 3).

The corrole dimers **2**–**4** (Figure 3) were able to reduce, at the same concentration and light conditions, ca. 6.5 log_10_ CFU/mL (dimer **3**), 5.1 log_10_ CFU/mL (corrole dimer **4**), and 2.4 log_10_ CFU/mL (corrole dimer **2**) after 180 min of irradiation. Despite the differences observed in the inactivation ratio between dimers, the inactivation efficiency between corrole dimers **3** and **4** was not enough to be considered as a significant difference along the treatment time (Anova, *p* > 0.05). The differences between the inactivation of corrole dimers **4** and **2** were also not significantly different until the 150 min treatment time (Anova, *p* > 0.05), and were only different after 180 min of aPDT treatment (Anova, *p* < 0.05). However, the inactivation efficiency between corrole dimers **3** and **2** was significantly different along the treatment time (Anova, *p* < 0.05).

The viability of *S. aureus* cells was not affected either in the presence of any PS without light (DC), nor by light alone (LC) (Figure 3) (individual values of LC and DC for each PS are presented in Appendix A).

Considering that all corrole dimers presented a similar efficiency in generating ^1^O_2_, (although lower than of the monomeric precursor **1**), we hypothesized that the differences between dimer inactivation efficiency towards *S. aureus* could be related to compound uptakes, as there may be some binding constraint to the bacterial structures, especially associated to their chemical structures. Corrole dimer **2** is a highly fused conjugated dimeric corrole molecule, with two linkages between the corrole monomers that might present a higher constraint when compared to dimers **3** and **4**, which have only one linkage between the monomers, possibly justifying the low antimicrobial capacity of corrole dimer **2** (below 3.0 log photoinactivation after 180 min). The other corrole dimers with more flexible structures could have a higher affinity to the bacterial structures and a consequently higher inactivation rate.

In this way, we carried out the uptake assays (Figure 4) of all the compounds (corrole precursor **1**, and corrole dimers **2**, **3**, and **4**).

It was verified that despite slight differences among the PS molecules adhesion results, there were no significant differences in the adhesion of the monomeric corrole **1** and the corrole dimer **3** (Anova, *p* > 0.05). However, the adhesion of the corrole dimer **4** showed to be significantly different from the other compounds, presenting a much higher adhesion value to bacterial cells (Anova, *p* < 0.05). These data do not seem to be in agreement with the inactivation efficiency, since corrole dimers **4** and **3** did not show significant activity differences, and the higher uptake of dimer **4** to the bacterial cells was not reflected in its inactivation efficiency. It can be hypothesized that the high value found for dimer **4** adhesion to bacterial cells might be due to an aggregation phenomenon, leading to a significant increase of molecules present on the cell surface. However, when the phenomenon of aggregation of PS occurs, the production of singlet oxygen decreases [46,47], justifying the fact that the presence of a higher number of adhering molecules was not reflected in a high inactivation rate when compared with corrole dimer **3**, despite their similar behavior in singlet oxygen generation in DMF.

It is worth referring that the uptake procedure significantly altered the spectroscopic features of corrole dimer **2** (data not shown). This fact is probably due to some degradation occurring in the presence of the bacterial suspension or bacterial content, which might justify the weaker inactivation profile observed. For this reason, the uptake of dimer **2** was not determined.

The uptake results and the corresponding singlet oxygen generation ability allow the confirmation, once again, that microbial inactivation efficiency is influenced by factors as PS adhesion to the bacterial cells, its aggregation behavior, and its capacity to generate ROS, in particular singlet oxygen; therefore, the inactivation outcome does not depend solely on one of these factors, but the combination of these influencing the process in parallel. In a previous work, Cardote et al. [15] showed that corrole **1** (used at a concentration of 10 µM) was not able to efficiently inactivate the Gram(−) bacterium *A. fischeri*, even when a higher irradiance was used (100 mW/cm^2^). However, we demonstrated that neutral corroles can efficiently inactivate a Gram(+) bacterium at physiological pH without the requirement of membrane disruptors in order to facilitate the PS entrance on the bacterial cells [15,48,49,50].

### 3.5. PS Cytotoxicity

Keeping in mind the potential of the PSs herein prepared as antibacterial agents in the treatment of infections, the cytotoxicities of the promising PSs—corrole precursor **1** and corrole dimer **3**—were determined towards Vero cells derived from the kidney of an African green monkey (*Cercopithecus aethiops*). This cell line is one of the most common mammalian continuous cell lines used in research [51].

Vero cells were exposed to PSs **1** and **3** for 24 h according to ISO guidelines (ISO 10993-5) for in vitro cytotoxicity of medical devices. The cytotoxicity of DMSO at the same percentage used to prepare the PSs stock solutions for the aPDT assays was also evaluated. The results obtained are presented in Figure 5.

The results showed that corrole precursor **1** induces a decrease of 23.3% (*p* < 0.001) in the Vero cells’ viability, while corrole dimer **3** promotes a decrease of 11.7% (*p* < 0.01) in the cell line survival. DMSO at 3% did not induce any alteration in the Vero cell line survival. Since ISO guidelines refer specify that a material that promotes cell viability rates higher than 70% compared to the control group shall be considered non-cytotoxic [37], corrole precursor **1** and dimer **3** should be considered safe for clinical applications.

## 4. Conclusions

This study describes a new synthetic route providing direct access to gallium(III) corrole dimers complexes and their potential application as antibacterial agents. The results show that all the dimers exhibit the ability to photoinactivate the Gram(+) bacterium *S. aureus* under white light irradiation, although their photodynamic efficiency is dependent on their position, the type of linkage between the monomer units, and aggregation behavior. Nevertheless, the one-pot reaction created PS corrole dimers able to reduce 99.9999% (decrease of 5 log CFU/mL) of the *S. aureus* strain. It is envisaged that further modifications in corrole dimeric structures will allow the aggregation phenomena in an aqueous medium to be overcome. Corrole precursor **1** and corrole dimer **3** were shown to be effective antibacterial PSs that were non-cytotoxic for Vero cell lines, attesting to their safety in clinical usage. This study demonstrates that corrole dimers are good photosensitizers for killing bacterium such as *S. aureus*.

## Data Availability

Not applicable.

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
