# Peer review of "Can Corrole Dimers Be Good Photosensitizers to Kill Bacteria?"

_microorganisms, 2022, doi:10.3390/microorganisms10061167_

Round 1
Reviewer 1 Report
It was a pleasure for me to review the manuscript of Lacerda et al. as we are working in a similar research field. The manuscript "Can corrole dimers be good photosensitizers to kill bacteria?" for the first time, described that corrole dimers are good photosensitizers for aPDT, which can kill pathogenic bacteria such as S. aureus. In terms of increasing bacterial infections worldwide, the paper is highly relevant to the scientific community. Therefore, I recommend the manuscript for publication after some minor revisions as indicated below:
Line 34: their optical - ,their optical
Line 37: namely for cancer Photo - , namely for cancer Photo
Line 41: in the presence of dioxygen- oxygen?
Line 56: of non-charged corrole - electrically neutral?
Line 61: ability to generate singlet oxygen - only singlet oxygen or other ROS types also?
Line 97: Bacterial stocks of the bacterium – Stocks of the bacterium
Line 101: until the stationary phase of the bacterial culture is reached – which optical density it was, and why did you choose the stationary phase?
Line 111: Light source conditions: What was the distance from the LED to the sample?
Line 125: Light controls containing only bacterial suspension of S. aureus in PBS - Did you check DMSO dark toxicity? If so, then it should describe in the methods.
Line 317: Which ANOVA test did you use?
Line 318: (> 8.0 log10 CFU/mL reduction) - It isn't correct to show 0 logs on the figure because your method has a detection limit. What is the detection limit of the used method?
Line 319: light dose 0.45 kJ/cm2 – How did you calculate the light dose? Why did you use kJ/cm2, not mJ/cm2?
Line 333: Maybe it's better to use it in the X-axis illumination dose, not the time? Also, it isn't correct to show dark control on the same figure axis. Moreover, you presented it in supplements.
Line 345: we hypothesized that the differences- Could differences between dimers' inactivation efficiency towards S. aureus be related also to the ability to produce other ROS (not only singlet oxygen)?
Line 365: dimer 4 adhesion to bacterial cells is due to an aggregation phenomenon- Have you checked spectrophotometrically the degree of aggregation of your dimers, such as Triton X-100 detergent?
Author Response
Response to Reviewer 1
It was a pleasure for me to review the manuscript of Lacerda et al. as we are working in a similar research field. The manuscript "Can corrole dimers be good photosensitizers to kill bacteria?" for the first time, described that corrole dimers are good photosensitizers for aPDT, which can kill pathogenic bacteria such as S. aureus. In terms of increasing bacterial infections worldwide, the paper is highly relevant to the scientific community. Therefore, I recommend the manuscript for publication after some minor revisions as indicated below:
We would like to thank the reviewer for recognizing the value of the work and for the comments that by certain would improve the quality of the work.
Line 34: their optical - ,their optical
Line 37: namely for cancer Photo - , namely for cancer Photo
As suggested, the commas were added to the text.
Line 41: in the presence of dioxygen- oxygen?
We understand the concern of the reviewer since the designation of dioxygen in alternative to molecular oxygen (O2) is based on the most recent IUPAC recommendations for inorganic nomenclature (red book). According with these recommendations, where it is referred molecular oxygen should be named as dioxygen. So, we opt to introduce the most recent IUPAC recommendation in the manuscript but considering that the designation is not yet well known we decided in the revised manuscript to add the molecular formula (3O2) after the word dioxygen.
Line 56: of non-charged corrole - electrically neutral?
Yes, in fact the designation of non-charged corroles is similar to the one suggested to the reviewer, of electrically neutral corroles. This information was added to the new version of the manuscript.
Line 61: ability to generate singlet oxygen - only singlet oxygen or other ROS types also?
We thank the reviewer for this alert since other ROS could be involved in the photodynamic action. However, it is well-known that in the photodynamic mechanism
mediated by tetrapyrrolic macrocycles the main cytotoxic species is singlet oxygen. So, in order to avoid misunderstandings in the revised version we change the comment to: “Among the key photophysical and photochemical characteristics as ability to generate reactive oxygen species (ROS), in particular singlet oxygen.”
Line 97: Bacterial stocks of the bacterium – Stocks of the bacterium
The authors thank for the correction and the sentence was changed according to the reviewer suggestion.
Line 101: until the stationary phase of the bacterial culture is reached – which optical density it was, and why did you choose the stationary phase?
According with previous studies, it is recognised that the susceptibility of bacteria to antibiotic drugs is greatly dependent on the growth rate of the bacteria, being more susceptible during the exponential growth phase (Eng et al. 1991; Li ate al. 2016; Li et al. 2017). As so, in our group we choose to test the efficiency of antimicrobials during the less susceptible growth phase, after reaching the stationary growth phase, when the metabolic processes are slower and consequently more demanding in terms of evaluating the efficiency of an antimicrobial approach. Regarding Staphylococcus aureus and the conditions used in our study, the stationary growth phase is reached after 16 – 18 h of growth. After this time of incubation, a bacterial concentration between 108-109 CFU/mL, corresponding to OD (600 nm) of ca. 1.6, is obtained.
Some of this information was added to the revised version of the manuscript.
Li, J., Xie, S., Ahmed, S., Wang, F., Gu, Y., Zhang, C., Chai, X., Wu, Y., Cai, J., & Cheng, G. (2017). Antimicrobial Activity and Resistance: Influencing Factors. Frontiers in pharmacology, 8(364). Doi: 10.3389/fphar.2017.00364
Li, B.; Qiu, Y.; Shia, H.; & Yin H. (2016). The importance of lag time extension in determining bacterial resistance to antibiotics. Analyst, 141(3059). Doi: 10.1039/c5an02649k
Eng, R. H. K.; Padberg, F. T.; Smith, S. M.; Tan, E. N.; Cherubin, C. E. (1991). Bactericidal Effects of Antibiotics on Slowly Growing and Nongrowing Bacteria. Antimicrobial Agents and Chemotherapy, 35(9).
Line 111: Light source conditions: What was the distance from the LED to the sample?
We understand this comment since the LED power is strongly dependent on the distance between the LED source and sample. In order to avoid alterations on the irradiation
conditions, in all the assays and before each experiment, the light irradiance was checked using a power meter FieldMaxII-TOP combined with a high-sensitivity thermopile sensor PS19Q (Coherent, Santa Clara, USA) as it is mentioned in the article. Nevertheless, the distance between the LED and the sample was around 3.0 cm and this information was added in the revised version of the manuscript.
Line 125: Light controls containing only bacterial suspension of S. aureus in PBS - Did you check DMSO dark toxicity? If so, then it should describe in the methods.
Before the aPDT experiments, toxicity tests of the PSs solvent DMSO were performed. Their description and results are presented on the Supplementary Material, Figure S2. However, taking in account this pertinent comment, we added the information about these experiments on the Materials and Methods section of the manuscript – section “2.2.4 aPDT experimental setup”, as following: “All the aPDT assays were performed at the same PS concentration (15 μM). The concentrations of the PSs were chosen according to the previously performed toxicity tests of the solvent (DMSO) considering the concentration of the PSs stock solutions (500 μM).”
Also, a procedure description was added to the Supplementary Material in a new methods section “S5. Determination of DMSO toxicity in S. aureus cells”.
Line 317: Which ANOVA test did you use?
The Tukey’s multiple comparisons test was performed, as described in the Materials and Methods section “2.4 Statistical analysis”, since this statistical test allowed us to compare all possible pairs of means among every treatment.
Line 318: (> 8.0 log10 CFU/mL reduction) - It isn't correct to show 0 logs on the figure because your method has a detection limit. What is the detection limit of the used method?
The authors thank the reviewer for this note. The detection limit of the method used is 2.0 log CFU/mL. This information has been added to the Materials and Methods, in the section “2.2.4 aPDT experimental setup”. The same consideration was also added to the Figure 3, by complementing the graph with a dashed line which points out the detection limit of the method. The information “The dashed line represents the detection limit of the method used.” was also added to the figure’ legend.
Line 319: light dose 0.45 kJ/cm2 – How did you calculate the light dose? Why did you use kJ/cm2, not mJ/cm2?
The light dose was calculated by the equation irradiance (W/cm2) = energy (J/cm2)/ time (s). In general, the selection of the light dose unit is dependent on the authors and in here
we consider the same unit used in our previously published papers. However, in order to clarify this aspect, the authors change the light dose to the SI base units and 0.45 kJ/cm2 was replaced by 450 J/cm2.
Line 333: Maybe it's better to use it in the X-axis illumination dose, not the time? Also, it isn't correct to show dark control on the same figure axis. Moreover, you presented it in supplements.
According to the reviewer suggestion, in the new version, the authors included the light dose for each experimental set in the X axis.
Also, in accordance with the suggestion, the dark control was eliminated from the graph, and a note was added to the legend of the figure.
Line 345: we hypothesized that the differences- Could differences between dimers' inactivation efficiency towards S. aureus be related also to the ability to produce other ROS (not only singlet oxygen)?
We appreciate the reviewer comment. We agree that singlet oxygen might not be the only ROS produced during the photodynamic treatment as it is well documented by our and other groups for tetrapyrrolic derivatives and analogues. However, it is recognised that the photodynamic efficiency of this type of compounds towards bacterial cells is in general strongly dependent on singlet oxygen production and in its absence the aPDT treatment would be strongly affected. In fact, the possibility to be related to a different generation of other ROS cannot be totally eliminated, although considering the similarity of the structures of the dimers, we believe that their contribution would be also similar as happen with their production of singlet oxygen
This information was added to the discussion of the revised version.
Line 365: dimer 4 adhesion to bacterial cells is due to an aggregation phenomenon- Have you checked spectrophotometrically the degree of aggregation of your dimers, such as Triton X-100 detergent?
In fact, we did not check spectrophotometrically the degree of aggregation of the dimers. Since we hypothesized that an aggregation phenomenon could occur since we obtained a higher uptake value of dimer 4 we rephrased the sentence and instead of “It can be hypothesized that the high value found for dimer 4 adhesion to bacterial cells is due to an aggregation phenomenon, leading to a significant increase of molecules present in the cell surface.” we changed to “… might be due…”.
Reviewer 2 Report
The work presents the investigation on corrole dimers as new potential photosensitizers for aPDT. Generally, the paper is well constructured, the aim, methodology and results are clearly presented. There are only two issues that, in my opinion, should be taken into account:
- have authors tried to compare photosensitizing abilities of dimers, i.e. ROS production, under white light? This could be done for example with TPCPD trap and UV-Vis.
- have authors tried to exclude formation of other types of ROS under light illumination?
Author Response
Responses to Reviewer2
The work presents the investigation on corrole dimers as new potential photosensitizers for aPDT. Generally, the paper is well constructured, the aim, methodology and results are clearly presented. There are only two issues that, in my opinion, should be taken into account:
We would like to thank the reviewer for recognizing the value of the work and for the comments that by certain will improve the quality of the work.
1. have authors tried to compare photosensitizing abilities of dimers, i.e., ROS production, under white light? This could be done for example with TPCPD trap and UV-Vis.
Answer: We appreciate the reviewer concerns. In fact, we tried to compare the photosensitizing ability of the dimers through the singlet oxygen determination for each dimer. However, to achieve this goal is important to fix the number of photons for all molecules; regarding this, the dimers and the reference were exposed to a monochromatic light (420 nm) at the same O.D. to fix the number of photons. Under these conditions, it was possible to assess their ability to produce 1O2 and then to calculate their singlet oxygen quantum yield according with the information available in SM (section S4). For these types of assays, we use an indirect method, where 9,10-dimethylantracene (DMA) was selected as a reporter of 1O2 generation due to its recognized efficiency to quench this ROS species (Gunther et al. 2000; Santos et al. 2021). Moreover, since the compounds have different absorption features (different wavelength absorption in the visible range and different molar absorption coefficients), the use of white light would not allow to compare these results.
Gunther G., E. Lemp, A. L. Zanocco. 2000. On the use of 9,10-dimethylanthracene as chemical rate constant actinometer in singlet molecular oxygen reactions. Boletin de la Sociedad Chilena de Quimica. 45(4). Doi: 10.4067/S0366-16442000000400018
Santos I., Gamelas S.R.D., Vieira C., Faustino M.A.F., et al.. 2021.Pyrazole-pyridinium porphyrins and chlorins as powerful photosensitizers for photoinactivation of planktonic and biofilm forms of E. coli. Dyes and Pigments. 109557 (193) Doi:10.1016/j.dyepig.2021.109557
2. have authors tried to exclude formation of other types of ROS under light illumination?
Answer: The aim of this work was to assess the potential of dimeric corroles to act as photosensitizer agents. Based on the photodynamic action of tetrapyrrolic macrocycles which is in general mediated by their ability to generate singlet oxygen, in this work we only evaluated their efficiency to generate singlet oxygen. However, the contribution of
other type of ROS cannot be excluded and in the new version we try to contemplate that possibility.
Following the suggestion, text was added to the discussion section, beginning in line 313: “Singlet oxygen might not be the only ROS produced during the photodynamic treatment as it is well documented by our and other groups for the tetrapyrrolic derivatives and analogues. However, it is recognized that the photodynamic efficiency of this type of compounds towards bacterial cells is in general strongly dependent on singlet oxygen production and in its absence the aPDT treatment would be strongly affected. In fact, the possibility to be related to a different generation of other ROS cannot be eliminated, although considering the similitude of the structures of the dimers we believe that their contribution would be also similar as happen with their production of singlet oxygen.”
Round 2
Reviewer 2 Report
Authors have addressed all my questions, thus I suggest acceptance of the manuscript.